# Compression with Flows via Local Bits-Back Coding

**Jonathan Ho**
UC Berkeley
jonathanho@berkeley.edu

**Evan Lohn**
UC Berkeley
evan.lohn@berkeley.edu

**Pieter Abbeel**
UC Berkeley, covariant.ai
pabbeel@cs.berkeley.edu

## Abstract

Likelihood-based generative models are the backbones of lossless compression due to the guaranteed existence of codes with lengths close to negative log likelihood. However, there is no guaranteed existence of computationally efficient codes that achieve these lengths, and coding algorithms must be hand-tailored to specific types of generative models to ensure computational efficiency. Such coding algorithms are known for autoregressive models and variational autoencoders, but not for general types of flow models. To fill in this gap, we introduce local bits-back coding, a new compression technique for flow models. We present efficient algorithms that instantiate our technique for many popular types of flows, and we demonstrate that our algorithms closely achieve theoretical codelengths for state-of-the-art flow models on high-dimensional data.

## 1 Introduction

To devise a lossless compression algorithm means to devise a uniquely decodable code whose expected length is as close as possible to the entropy of the data. A general recipe for this is to first train a generative model by minimizing cross entropy to the data distribution, and then construct a code that achieves lengths close to the negative log likelihood of the model. This recipe is justified by classic results in information theory that ensure that the second step is possible—in other words, optimizing cross entropy optimizes the performance of some hypothetical compressor. And, thanks to recent advances in deep likelihood-based generative models, these hypothetical compressors are quite good. Deep autoregressive models, latent variable models, and flow models are now achieving state-of-the-art cross entropy scores on a wide variety of real-world datasets in speech, videos, text, images, and other domains [45, 46, 39, 34, 5, 31, 6, 22, 10, 11, 23, 35, 19, 44, 25, 29].

But we are not interested in hypothetical compressors. We are interested in practical, computationally efficient compressors that scale to high-dimensional data and harness the excellent cross entropy scores of modern deep generative models. Unfortunately, naively applying existing codes, like Huffman coding [21], requires computing the model likelihood for all possible values of the data, which expends computational resources scaling exponentially with the data dimension. This inefficiency stems from the lack of assumptions about the generative model's structure.

Coding algorithms must be tailored to specific types of generative models if we want them to be efficient. There is already a rich literature of tailored coding algorithms for autoregressive models and variational autoencoders built from conditional distributions which are already tractable for coding [38, 12, 18, 13, 42], but there are currently no such algorithms for general types of flow models [32]. It seems that this lack of efficient coding algorithms is a con of flow models that stands at odds with their many pros, like fast and realistic sampling, interpretable latent spaces, fast likelihood evaluation, competitive cross entropy scores, and ease of training with unbiased log likelihood gradients [10, 11, 23, 19].

To rectify this situation, we introduce *local bits-back coding*, a new technique for turning a general, pretrained, off-the-shelf flow model into an efficient coding algorithm suitable for continuous data discretized to high precision. We show how to implement local bits-back coding without assumptions

on the flow structure, leading to an algorithm that runs in polynomial time and space with respect to the data dimension. Going further, we show how to tailor our implementation to various specific types of flows, culminating in a fully parallelizable algorithm for RealNVP-type flows that runs in linear time and space with respect to the data dimension and is fully parallelizable for both encoding and decoding. We then show how to adapt local bits-back coding to losslessly code data discretized to arbitrarily low precision, and in doing so, we obtain a new compression interpretation of dequantization, a method commonly used to train flow models on discrete data. We test our algorithms on recently proposed flow models trained on real-world image datasets, and we find that they are computationally efficient and attain codelengths in close agreement with theoretical predictions. Open-source code is available at `https://github.com/hojonathanho/localbitsback`.

## 2 Preliminaries

**Lossless compression**   We begin by defining lossless compression of $d$-dimensional discrete data $\mathbf{x}^\circ$ using a probability mass function $p(\mathbf{x}^\circ)$ represented by a generative model. It means to construct a uniquely decodable code $C$, which is an injective map from data sequences to binary strings, whose lengths $|C(\mathbf{x}^\circ)|$ are close to $-\log p(\mathbf{x}^\circ)$ [7].[1] The rationale is that if the generative model is expressive and trained well, its cross entropy will be close to the entropy of the data distribution. So, if the lengths of $C$ match the model's negative log probabilities, the expected length of $C$ will be small, and hence $C$ will be a good compression algorithm. Constructing such a code is always possible in theory because the Kraft-McMillan inequality [27, 30] ensures that there always exists some code with lengths $|C(\mathbf{x}^\circ)| = \lceil -\log p(\mathbf{x}^\circ) \rceil \approx -\log p(\mathbf{x}^\circ)$.

**Flow models**   We wish to construct a computationally efficient code specialized to a flow model $f$, which is a differentiable bijection between continuous data $\mathbf{x} \in \mathbb{R}^d$ and latents $\mathbf{z} = f(\mathbf{x}) \in \mathbb{R}^d$ [9–11]. A flow model comes with a density $p(\mathbf{z})$ on the latent space and thus has an associated sampling process—$\mathbf{x} = f^{-1}(\mathbf{z})$ for $\mathbf{z} \sim p(\mathbf{z})$—under which it defines a probability density function via the change-of-variables formula for densities:

$$-\log p(\mathbf{x}) = -\log p(\mathbf{z}) - \log |\det \mathbf{J}(\mathbf{x})| \tag{1}$$

where $\mathbf{J}(\mathbf{x})$ denotes the Jacobian of $f$ at $\mathbf{x}$. Flow models are straightforward to train with maximum likelihood, as Eq. (1) allows unbiased exact log likelihood gradients to be computed efficiently.

**Dequantization**   Standard datasets such as CIFAR10 and ImageNet consist of discrete data $\mathbf{x}^\circ \in \mathbb{Z}^d$. To make a flow model suitable for such discrete data, it is standard practice to define a derived discrete model $P(\mathbf{x}^\circ) := \int_{[0,1)^d} p(\mathbf{x}^\circ + \mathbf{u}) \, d\mathbf{u}$ to be trained by minimizing a dequantization objective, which is a variational bound on the codelength of $P(\mathbf{x}^\circ)$:

$$\mathbb{E}_{\mathbf{u} \sim q(\mathbf{u}|\mathbf{x}^\circ)} \left[ -\log \frac{p(\mathbf{x}^\circ + \mathbf{u})}{q(\mathbf{u}|\mathbf{x}^\circ)} \right] \geq -\log \int_{[0,1)^d} p(\mathbf{x}^\circ + \mathbf{u}) \, d\mathbf{u} = -\log P(\mathbf{x}^\circ) \tag{2}$$

Here, $q(\mathbf{u}|\mathbf{x}^\circ)$ proposes dequantization noise $\mathbf{u} \in [0,1)^d$ that transforms discrete data $\mathbf{x}^\circ$ into continuous data $\mathbf{x}^\circ + \mathbf{u}$; it can be fixed to either a uniform distribution [43, 41, 39] or to another parameterized flow to be trained jointly with $f$ [19]. This dequantization objective serves as a theoretical codelength for flow models trained on discrete data, just like negative log probability mass serves as a theoretical codelength for discrete generative models [41].

## 3 Local bits-back coding

Our goal is to develop computationally efficient coding algorithms for flows trained with dequantization (2). In Sections 3.1 to 3.4, we develop algorithms that use flows to code continuous data discretized to high precision. In Section 3.5, we adapt these algorithms to losslessly code data discretized to low precision, attaining our desired codelength (2) for discrete data.

### 3.1 Coding continuous data using discretization

We first address the problem of developing coding algorithms that attain codelengths given by negative log densities of flow models, such as Eq. (1). Probability density functions do not directly map to codelength, unlike probability mass functions which enjoy the result of the Kraft-McMillan inequality. So, following standard procedure [7, section 8.3], we discretize the data to a high precision $k$

and code this discretized data with a certain probability mass function derived from the density model. Specifically, we tile $\mathbb{R}^d$ with hypercubes of volume $\delta_x := 2^{-kd}$; we call each hypercube a bin. For $\mathbf{x} \in \mathbb{R}^d$, let $B(\mathbf{x})$ be the unique bin that contains $\mathbf{x}$, and let $\bar{\mathbf{x}}$ be the center of the bin $B(\mathbf{x})$. We call $\bar{\mathbf{x}}$ the discretized version of $\mathbf{x}$. For a sufficiently smooth probability density function $p(\mathbf{x})$, such as a density coming from a neural network flow model, the probability mass function $P(\bar{\mathbf{x}}) := \int_{B(\bar{\mathbf{x}})} p(\mathbf{x}) \, d\mathbf{x}$ takes on the pleasingly simple form $P(\bar{\mathbf{x}}) \approx p(\bar{\mathbf{x}})\delta_x$ when the precision $k$ is large. Now we invoke the Kraft-McMillan inequality, so the theoretical codelength for $\bar{\mathbf{x}}$ using $P$ is

$$- \log P(\bar{\mathbf{x}}) \approx - \log p(\bar{\mathbf{x}})\delta_x \tag{3}$$

bits. This is the compression interpretation of the negative log density: it is a codelength for data discretized to high precision, when added to the total number of bits of discretization precision. It is this codelength, Eq. (3), that we will try to achieve with an efficient algorithm for flow models. We defer the problem of coding data discretized to low precision to Section 3.5.

## 3.2 Background on bits-back coding

The main tool we will employ is bits-back coding [47, 18, 13, 20], a coding technique originally designed for latent variable models (the connection to flow models is presented in Section 3.3 and is new to our work). Bits-back coding codes $\mathbf{x}$ using a distribution of the form $p(\mathbf{x}) = \sum_{\mathbf{z}} p(\mathbf{x}, \mathbf{z})$, where $p(\mathbf{x}, \mathbf{z}) = p(\mathbf{x}|\mathbf{z})p(\mathbf{z})$ includes a latent variable $\mathbf{z}$; it is relevant when $\mathbf{z}$ ranges over an exponentially large set, which makes it intractable to code with $p(\mathbf{x})$ even though coding with $p(\mathbf{x}|\mathbf{z})$ and $p(\mathbf{z})$ may be tractable individually. Bits-back coding introduces a new distribution $q(\mathbf{z}|\mathbf{x})$ with tractable coding, and the encoder jointly encodes $\mathbf{x}$ along with $\mathbf{z} \sim q(\mathbf{z}|\mathbf{x})$ via these steps:

1. Decode $\mathbf{z} \sim q(\mathbf{z}|\mathbf{x})$ from an auxiliary source of random bits
2. Encode $\mathbf{x}$ using $p(\mathbf{x}|\mathbf{z})$
3. Encode $\mathbf{z}$ using $p(\mathbf{z})$

The first step, which decodes $\mathbf{z}$ from random bits, produces a sample $\mathbf{z} \sim q(\mathbf{z}|\mathbf{x})$. The second and third steps transmit $\mathbf{z}$ along with $\mathbf{x}$. At decoding time, the decoder recovers $(\mathbf{x}, \mathbf{z})$, then recovers the bits the encoder used to sample $\mathbf{z}$ using $q$. So, the encoder will have transmitted extra information in addition to $\mathbf{x}$—precisely $\mathbb{E}_{\mathbf{z} \sim q(\mathbf{z}|\mathbf{x})} [- \log q(\mathbf{z}|\mathbf{x})]$ bits on average. Consequently, the net number of bits transmitted regarding $\mathbf{x}$ only will be $\mathbb{E}_{\mathbf{z} \sim q(\mathbf{z}|\mathbf{x})} [\log q(\mathbf{z}|\mathbf{x}) - \log p(\mathbf{x}, \mathbf{z})]$, which is redundant compared to the desired length $- \log p(\mathbf{x})$ by an amount equal to the KL divergence $D_{\mathrm{KL}} \left( q(\mathbf{z}|\mathbf{x}) \, \| \, p(\mathbf{z}|\mathbf{x}) \right)$ from $q$ to the true posterior.

Bits-back coding also works with continuous $\mathbf{z}$ discretized to high precision with negligible change in codelength [18, 42]. In this case, $q(\mathbf{z}|\mathbf{x})$ and $p(\mathbf{z})$ are probability density functions. Discretizing $\mathbf{z}$ to bins $\bar{\mathbf{z}}$ of small volume $\delta_z$ and defining the probability mass functions $Q(\bar{\mathbf{z}}|\mathbf{x})$ and $P(\bar{\mathbf{z}})$ by the method in Section 3.1, we see that the bits-back codelength remains approximately unchanged:

$$\mathbb{E}_{\bar{\mathbf{z}} \sim Q(\bar{\mathbf{z}}|\mathbf{x})} \left[ - \log \frac{p(\mathbf{x}|\bar{\mathbf{z}})P(\bar{\mathbf{z}})}{Q(\bar{\mathbf{z}}|\mathbf{x})} \right] \approx \mathbb{E}_{\bar{\mathbf{z}} \sim Q(\bar{\mathbf{z}}|\mathbf{x})} \left[ - \log \frac{p(\mathbf{x}|\bar{\mathbf{z}})p(\bar{\mathbf{z}})\delta_z}{q(\bar{\mathbf{z}}|\mathbf{x})\delta_z} \right] \approx \mathbb{E}_{\mathbf{z} \sim q(\mathbf{z}|\mathbf{x})} \left[ - \log \frac{p(\mathbf{x}|\mathbf{z})p(\mathbf{z})}{q(\mathbf{z}|\mathbf{x})} \right] \tag{4}$$

When bits-back coding is applied to a particular latent variable model, such as a VAE, the distributions involved may take on a certain meaning: $p(\mathbf{z})$ would be the prior, $p(\mathbf{x}|\mathbf{z})$ would be the decoder network, and $q(\mathbf{z}|\mathbf{x})$ would be the encoder network [25, 36, 8, 4, 13, 42, 26]. However, it is important to note that these distributions do not need to correspond explicitly to parts of the model at hand. Any will do for coding data losslessly, though some choices result in better codelengths. We exploit this fact in Section 3.3, where we apply bits-back coding to flow models by constructing artificial distributions $p(\mathbf{x}|\mathbf{z})$ and $q(\mathbf{z}|\mathbf{x})$, which do not come with a flow model by default.

## 3.3 Local bits-back coding

We now present *local bits-back coding*, our new high-level principle for using a flow model $f$ to code data discretized to high precision. Following Section 3.1, we discretize continuous data $\mathbf{x}$ into $\bar{\mathbf{x}}$, which is the center of a bin of volume $\delta_x$. The codelength we desire for $\bar{\mathbf{x}}$ is the negative log density of $f$ (1), plus a constant depending on the discretization precision:

$$- \log p(\bar{\mathbf{x}})\delta_x = - \log p(f(\bar{\mathbf{x}})) - \log |\det \mathbf{J}(\bar{\mathbf{x}})| - \log \delta_x \tag{5}$$

where $\mathbf{J}(\bar{\mathbf{x}})$ is the Jacobian of $f$ at $\bar{\mathbf{x}}$. We will construct two densities $\tilde{p}(\mathbf{z}|\mathbf{x})$ and $\tilde{p}(\mathbf{x}|\mathbf{z})$ such that bits-back coding attains Eq. (5). We need a small scalar parameter $\sigma > 0$, with which we define

$$\tilde{p}(\mathbf{z}|\mathbf{x}) := \mathcal{N}(\mathbf{z}; f(\mathbf{x}), \sigma^2 \mathbf{J}(\mathbf{x})\mathbf{J}(\mathbf{x})^\top) \qquad \text{and} \qquad \tilde{p}(\mathbf{x}|\mathbf{z}) := \mathcal{N}(\mathbf{x}; f^{-1}(\mathbf{z}), \sigma^2 \mathbf{I}) \tag{6}$$

To encode $\bar{\mathbf{x}}$, local bits-back coding follows the method described in Section 3.2 with continuous $\mathbf{z}$:

1. Decode $\bar{\mathbf{z}} \sim \tilde{P}(\bar{\mathbf{z}}|\mathbf{x}) = \int_{B(\bar{\mathbf{z}})} \tilde{p}(\mathbf{z}|\mathbf{x}) \, d\mathbf{z} \approx \tilde{p}(\bar{\mathbf{z}}|\mathbf{x})\delta_z$ from an auxiliary source of random bits
2. Encode $\bar{\mathbf{x}}$ using $\tilde{P}(\bar{\mathbf{x}}|\bar{\mathbf{z}}) = \int_{B(\bar{\mathbf{x}})} \tilde{p}(\mathbf{x}|\bar{\mathbf{z}}) \, d\mathbf{z} \approx \tilde{p}(\bar{\mathbf{x}}|\bar{\mathbf{z}})\delta_x$
3. Encode $\bar{\mathbf{z}}$ using $P(\bar{\mathbf{z}}) = \int_{B(\bar{\mathbf{z}})} p(\mathbf{z}) \, d\mathbf{z} \approx p(\bar{\mathbf{z}})\delta_z$

The conditional density $\tilde{p}(\mathbf{z}|\mathbf{x})$ (6) is artificially injected noise, scaled by $\sigma$ (the flow model $f$ remains unmodified). It describes how a local linear approximation of $f$ would behave if it were to act on a small Gaussian around $\bar{\mathbf{x}}$.

To justify local bits-back coding, we simply calculate its expected codelength. First, our choices of $\tilde{p}(\mathbf{z}|\mathbf{x})$ and $\tilde{p}(\mathbf{x}|\mathbf{z})$ (6) satisfy the following equation:

$$\mathbb{E}_{\mathbf{z} \sim \tilde{p}(\mathbf{z}|\mathbf{x})} \left[\log \tilde{p}(\mathbf{z}|\mathbf{x}) - \log \tilde{p}(\mathbf{x}|\mathbf{z})\right] = -\log|\det \mathbf{J}(\mathbf{x})| + O(\sigma^2) \tag{7}$$

Next, just like standard bits-back coding (4), local bits-back coding attains an expected codelength close to $\mathbb{E}_{\mathbf{z} \sim \tilde{p}(\mathbf{z}|\bar{\mathbf{x}})} L(\bar{\mathbf{x}}, \mathbf{z})$, where

$$L(\mathbf{x}, \mathbf{z}) := \log \tilde{p}(\mathbf{z}|\mathbf{x})\delta_z - \log \tilde{p}(\mathbf{x}|\mathbf{z})\delta_x - \log p(\mathbf{z})\delta_z \tag{8}$$

Equations (6) to (8) imply that the expected codelength matches our desired codelength (5), up to first order in $\sigma$ (see Appendix A for details):

$$\mathbb{E}_{\mathbf{z}} L(\mathbf{x}, \mathbf{z}) = -\log p(\mathbf{x})\delta_x + O(\sigma^2) \tag{9}$$

Note that local bits-back coding exactly achieves the desired codelength for flows (5), up to first order in $\sigma$. This is in stark contrast to bits-back coding with latent variable models like VAEs, for which the bits-back codelength is the negative evidence lower bound, which is redundant by an amount equal to the KL divergence from the approximate posterior to the true posterior [25].

Local bits-back coding always codes $\bar{\mathbf{x}}$ losslessly, no matter the setting of $\sigma$, $\delta_x$, and $\delta_z$. However, $\sigma$ must be small for the $O(\sigma^2)$ inaccuracy in Eq. (9) to be negligible. But for $\sigma$ to be small, the discretization volumes $\delta_z$ and $\delta_x$ must be small too, otherwise the discretized Gaussians $\tilde{p}(\bar{\mathbf{z}}|\mathbf{x})\delta_z$ and $\tilde{p}(\bar{\mathbf{x}}|\mathbf{z})\delta_x$ will be poor approximations of the original Gaussians $\tilde{p}(\mathbf{z}|\mathbf{x})$ and $\tilde{p}(\mathbf{x}|\mathbf{z})$. So, because $\delta_x$ must be small, the data $\mathbf{x}$ must be discretized to high precision. And, because $\delta_z$ must be small, a relatively large number of auxiliary bits must be available to decode $\bar{\mathbf{z}} \sim \tilde{p}(\bar{\mathbf{z}}|\mathbf{x})\delta_z$. We will resolve the high precision requirement for the data with another application of bits-back coding in Section 3.5, and we will explore the impact of varying $\sigma$, $\delta_x$, and $\delta_z$ on real-world data in experiments in Section 4.

### 3.4 Concrete local bits-back coding algorithms

We have shown that local bits-back coding attains the desired codelength (5) for data discretized to high precision. Now, we instantiate local bits-back coding with concrete algorithms.

#### 3.4.1 Black box flows

Algorithm 1 is the most straightforward implementation of local bits-back coding. It directly implements the steps in Section 3.3 by invoking an external procedure, such as automatic differentiation, to explicitly compute the Jacobian of the flow. It therefore makes no assumptions on the structure of the flow, and hence we call it the *black box* algorithm.

---

**Algorithm 1** Local bits-back encoding: for black box flows (decoding in Appendix B)

**Require:** data $\bar{\mathbf{x}}$, flow $f$, discretization volumes $\delta_x$, $\delta_z$, noise level $\sigma$
1: $\mathbf{J} \leftarrow \mathbf{J}_f(\bar{\mathbf{x}})$                            ▷ Compute the Jacobian of $f$ at $\bar{\mathbf{x}}$
2: Decode $\bar{\mathbf{z}} \sim \mathcal{N}(f(\bar{\mathbf{x}}), \sigma^2 \mathbf{J}\mathbf{J}^\top) \, \delta_z$           ▷ By converting to an AR model (Section 3.4.1)
3: Encode $\bar{\mathbf{x}}$ using $\mathcal{N}(f^{-1}(\bar{\mathbf{z}}), \sigma^2 \mathbf{I}) \, \delta_x$
4: Encode $\bar{\mathbf{z}}$ using $p(\bar{\mathbf{z}}) \, \delta_z$

---

Coding with $\tilde{p}(\mathbf{x}|\mathbf{z})$ (6) is efficient because its coordinates are independent [42]. The same applies to the prior $p(\mathbf{z})$ if its coordinates are independent or if another efficient coding algorithm already exists for it (see Section 3.4.3). Coding efficiently with $\tilde{p}(\mathbf{z}|\mathbf{x})$ relies on the fact that any multivariate Gaussian can be converted into a linear autoregressive model, which can be coded efficiently, one coordinate at a time, using arithmetic coding or asymmetric numeral systems. To see how, suppose $\mathbf{y} = \mathbf{J}\boldsymbol{\epsilon}$, where $\boldsymbol{\epsilon} \sim \mathcal{N}(\mathbf{0}, \mathbf{I})$ and $\mathbf{J}$ is a full-rank matrix (such as a Jacobian of a flow model). Let $\mathbf{L}$ be the Cholesky decomposition of $\mathbf{J}\mathbf{J}^\top$. Since $\mathbf{L}\mathbf{L}^\top = \mathbf{J}\mathbf{J}^\top$, the distribution of $\mathbf{L}\boldsymbol{\epsilon}$ is equal to the distribution of $\mathbf{J}\boldsymbol{\epsilon} = \mathbf{y}$, and so solutions $\tilde{\mathbf{y}}$ to the linear system $\mathbf{L}^{-1}\tilde{\mathbf{y}} = \boldsymbol{\epsilon}$ have the same distribution

as $\mathbf{y}$. Because $\mathbf{L}$ is triangular, $\mathbf{L}^{-1}$ is easily computable and also triangular, and thus $\tilde{\mathbf{y}}$ can be determined with back substitution: $\tilde{y}_i = (\epsilon_i - \sum_{j<i}(L^{-1})_{ij}\tilde{y}_j)/(L^{-1})_{ii}$, where $i$ increases from 1 to $d$. In other words, $p(\tilde{y}_i|\tilde{\mathbf{y}}_{<i}) \coloneqq \mathcal{N}(\tilde{y}_i; -L_{ii}\sum_{j<i}(L^{-1})_{ij}\tilde{y}_j, L_{ii}^2)$ is a linear autoregressive model that represents the same distribution as $\mathbf{y} = \mathbf{J}\epsilon$.

If nothing is known about the structure of the Jacobian of the flow, Algorithm 1 requires $O(d^2)$ space to store the Jacobian and $O(d^3)$ time to compute the Cholesky decomposition. This is certainly an improvement on the exponential space and time required by naive algorithms (Section 1), but it is still not efficient enough for high-dimensional data in practice. To make our coding algorithms more efficient, we need to make additional assumptions on the flow. If the Jacobian is always block diagonal, say with fixed block size $c \times c$, then the steps in Algorithm 1 can be modified to process each block separately in parallel, thereby reducing the required space and time to $O(cd)$ and $O(c^2 d)$, respectively. This makes Algorithm 1 efficient for flows that operate as elementwise transformations or as convolutions, such as activation normalization flows and invertible $1 \times 1$ convolution flows [23].

### 3.4.2 Autoregressive flows

An autoregressive flow $\mathbf{z} = f(\mathbf{x})$ is a sequence of one-dimensional flows $z_i = f_i(x_i; \mathbf{x}_{<i})$ for each coordinate $i \in \{1, \ldots, d\}$ [33, 24]. Algorithm 2 shows how to code with an autoregressive flow in linear time and space. It never explicitly calculates and stores the Jacobian of the flow, unlike Algorithm 1. Rather, it invokes one-dimensional local bits-back coding on one coordinate of the data at a time, thus exploiting the structure of the autoregressive flow in an essential way.

---

**Algorithm 2** Local bits-back encoding: for autoregressive flows (decoding in Appendix B)

**Require:** data $\bar{\mathbf{x}}$, autoregressive flow $f$, discretization volumes $\delta_x, \delta_z$, noise level $\sigma$
1: **for** $i = d, \ldots, 1$ **do**          ▷ Iteration ordering not mandatory, but convenient for ANS
2:      Decode $\bar{z}_i \sim \mathcal{N}(f_i(\bar{x}_i; \bar{\mathbf{x}}_{<i}), (\sigma f_i'(\bar{x}_i; \bar{\mathbf{x}}_{<i}))^2)\,\delta_z^{1/d}$      ▷ Neural net operations parallelizable over $i$
3:      Encode $\bar{x}_i$ using $\mathcal{N}(f_i^{-1}(\bar{z}_i; \bar{\mathbf{x}}_{<i}), \sigma^2)\,\delta_x^{1/d}$
4: **end for**
5: Encode $\bar{\mathbf{z}}$ using $p(\bar{\mathbf{z}})\,\delta_z$

---

A key difference between Algorithm 1 and Algorithm 2 is that the former needs to run the forward and inverse directions of the entire flow and compute and factorize a Jacobian, whereas the latter only needs to do so for each one-dimensional flow on each coordinate of the data. Consequently, Algorithm 2 runs $O(d)$ time and space, excluding resource requirements of the flow itself. The encoding procedure of Algorithm 2 resembles log likelihood computation for autoregressive flows, so the model evaluations it requires are completely parallelizable over data dimensions. The decoding procedure resembles sampling, so it requires $d$ model evaluations in serial (see Appendix B). These tradeoffs are entirely analogous to those of coding with discrete autoregressive models.

Autoregressive flows with further special structure lead to even more efficient implementations of Algorithm 2. As an example, let us focus on a NICE/RealNVP coupling layer [10, 11]. This type of flow computes $\mathbf{z}$ by splitting the coordinates of the input $\mathbf{x}$ into two halves, $\mathbf{x}_{\leq d/2}$, and $\mathbf{x}_{>d/2}$. The first half is passed through unchanged as $\mathbf{z}_{\leq d/2} = \mathbf{x}_{\leq d/2}$, and the second half is passed through an elementwise transformation $\mathbf{z}_{>d/2} = f(\mathbf{x}_{>d/2}; \mathbf{x}_{\leq d/2})$ which is conditioned on the first half. Specializing Algorithm 2 to this kind of flow allows both encoding and decoding to be parallelized over coordinates, resembling how the forward and inverse directions for inference and sampling can be parallelized for these flows [10, 11]. See Appendix B for the complete algorithm listing.

Algorithm 2 is not the only known efficient coding algorithm for autoregressive flows. For example, if $f$ is an autoregressive flow whose prior $p(\mathbf{z}) = \prod_i p_i(z_i)$ is independent over coordinates, then $f$ can be rewritten as a continuous autoregressive model $p(x_i|\mathbf{x}_{<i}) = p_i(f(x_i; \mathbf{x}_{<i}))|f'(x_i; \mathbf{x}_{<i})|$, which can be discretized and coded one coordinate at a time using arithmetic coding or asymmetric numeral systems. The advantage of Algorithm 2, as we will see next, is that it applies to more complex priors that prevent the distribution over $\mathbf{x}$ from naturally factorizing as an autoregressive model.

### 3.4.3 Compositions of flows

Flows like NICE, RealNVP, Glow, and Flow++ [10, 11, 23, 19] are composed of many intermediate flows: they have the form $f(\mathbf{x}) = f_K \circ \cdots \circ f_1(\mathbf{x})$, where each of the $K$ layers $f_i$ is one of the types of flows discussed above. These models derive their expressiveness from applying simple flows many times, resulting in a expressive composite flow. The expressiveness of the composite

flow suggests that coding will be difficult, but we can exploit the compositional structure to code efficiently. Since the composite flow $f = f_K \circ \cdots \circ f_1$ can be interpreted as a single flow $\mathbf{z}_1 = f_1(\mathbf{x})$ with a flow prior $f_K \circ \cdots \circ f_2(\mathbf{z}_1)$, all we have to do is code the first layer $f_1$ using the appropriate local bits-back coding algorithm, and when coding its output $\mathbf{z}_1$, we recursively invoke local bits-back coding for the prior $f_K \circ \cdots \circ f_2$ [26]. A straightforward inductive argument shows that this leads to the correct codelength. If coding any $\mathbf{z}_1$ with $f_K \circ \cdots \circ f_2$ achieves the expected codelength $-\log p_{f_K \circ \cdots \circ f_2}(\mathbf{z}_1)\delta_z + O(\sigma^2)$, then the expected codelength for $f_1$, using $f_K \circ \cdots \circ f_2$ as a prior, is $-\log p_{f_K \circ \cdots \circ f_2}(f_1(\mathbf{x})) - \log|\det \mathbf{J}_{f_1}(\mathbf{x})| - \log \delta_x + O(\sigma^2)$. Continuing the same into $f_K \circ \cdots \circ f_2$, we conclude that the resulting expected codelength

$$-\log p(\mathbf{z}_K) - \sum_{i=1}^{K} \log|\det \mathbf{J}_{f_i}(\mathbf{z}_{i-1})| - \log \delta_x + O(\sigma^2), \tag{10}$$

where $\mathbf{z}_0 := \mathbf{x}$, is what we expect from coding with the whole composite flow $f$. This codelength is averaged over noise injected into each layer $\mathbf{z}_i$, but we find that this is not an issue in practice. Our experiments in Section 4 show that it is easy to make $\sigma$ small enough to be negligible for neural network flow models, which are generally resistant to activation noise.

We call this the *compositional* algorithm. Its significance is that, provided that coding with each intermediate flow is efficient, coding with the composite flow is efficient too, despite the complexity of the composite flow as a function class. The composite flow's Jacobian never needs to be calculated or factorized, leading to dramatic speedups over using Algorithm 1 on the composite flow as a black box (Section 4). In particular, coding with RealNVP-type models needs just $O(d)$ time and space and is fully parallelizable over data dimensions.

## 3.5 Dequantization for coding unrestricted-precision data

We have shown how to code data discretized to high precision, achieving codelengths close to $-\log p(\bar{\mathbf{x}})\delta_x$. In practice, however, data is usually discretized to low precision; for example, images from CIFAR10 and ImageNet consist of integers in $\{0, 1, \ldots, 255\}$. Coding this kind of data directly would force us to code at a precision $-\log \delta_x$ much higher than 1, which would be a waste of bits.

To resolve this issue, we propose to use this extra precision within another bits-back coding scheme to arrive at a good lossless codelength for data at its original precision. Let us focus on the setting of coding integer-valued data $\mathbf{x}^\circ \in \mathbb{Z}^d$ up to bins of volume 1. Recall from Section 2 that flow models are trained on such data by minimizing a dequantization objective (2), which we reproduce here:

$$\mathbb{E}_{\mathbf{u} \sim q(\mathbf{u}|\mathbf{x}^\circ)} [\log q(\mathbf{u}|\mathbf{x}^\circ) - \log p(\mathbf{x}^\circ + \mathbf{u})] \tag{11}$$

Above, $q(\mathbf{u}|\mathbf{x}^\circ)$ is a dequantizer, which adds noise $\mathbf{u} \in [0, 1)^d$ to turn $\mathbf{x}^\circ$ into continuous data $\mathbf{x}^\circ + \mathbf{u}$ [43, 41, 39, 19]. We assume that the dequantizer is itself provided as a flow model, specified by $\mathbf{u} = q_{\mathbf{x}^\circ}(\boldsymbol{\epsilon}) \in [0, 1)^d$ for $\boldsymbol{\epsilon} \sim p(\boldsymbol{\epsilon})$, as in [19]. In Algorithm 3, we propose a bits-back coding scheme in which $\bar{\mathbf{u}} \sim q(\bar{\mathbf{u}}|\mathbf{x}^\circ)\delta_x$ is decoded from auxiliary bits using local bits-back coding, and $\mathbf{x}^\circ + \mathbf{u}$ is encoded using the original flow $p(\mathbf{x}^\circ + \bar{\mathbf{u}})\delta_x$, also using local bits-back coding.

---

**Algorithm 3** Local bits-back encoding with variational dequantization (decoding in Appendix B)

---
**Require:** discrete data $\mathbf{x}^\circ$, flow density $p$, dequantization flow conditional density $q$, discretization volume $\delta_x$
1: Decode $\bar{\mathbf{u}} \sim q(\bar{\mathbf{u}}|\mathbf{x}^\circ)\,\delta_x$ via local bits-back coding
2: $\bar{\mathbf{x}} \leftarrow \mathbf{x}^\circ + \bar{\mathbf{u}}$                                                   ▷ Dequantize
3: Encode $\bar{\mathbf{x}}$ using $p(\bar{\mathbf{x}})\,\delta_x$ via local bits-back coding

---

The decoder, upon receiving $\mathbf{x}^\circ + \bar{\mathbf{u}}$, recovers the original $\mathbf{x}^\circ$ and $\bar{\mathbf{u}}$ by rounding (see Appendix B for the full pseudocode). So, the net codelength for Algorithm 3 is given by subtracting the bits needed to decode $\bar{\mathbf{u}}$ from the bits needed to encode $\mathbf{x}^\circ + \bar{\mathbf{u}}$:

$$\log q(\bar{\mathbf{u}}|\mathbf{x}^\circ)\delta_x - \log p(\mathbf{x}^\circ + \bar{\mathbf{u}})\delta_x + O(\sigma^2) = \log q(\bar{\mathbf{u}}|\mathbf{x}^\circ) - \log p(\mathbf{x}^\circ + \bar{\mathbf{u}}) + O(\sigma^2) \tag{12}$$

This codelength closely matches the dequantization objective (11) on average, and it is reasonable for the low-precision discrete data $\mathbf{x}^\circ$ because, as we stated in Section 2, it is a variational bound on the codelength of a certain discrete generative model for $\mathbf{x}^\circ$, and modern flow models are explicitly trained to minimize this bound [43, 41, 19]. Since the resulting code is lossless for $\mathbf{x}^\circ$, Algorithm 3 provides a new compression interpretation of dequantization: it converts a code suitable for high precision data into a code suitable for low precision data, just as the dequantization objective (11) converts a model suitable for continuous data into a model suitable for discrete data [41].

# 4 Experiments

We designed experiments to investigate the following: (1) how well local bits-back codelengths match the theoretical codelengths of modern flow models on high-dimensional data, (2) the effects of the precision and noise parameters $\delta$ and $\sigma$ on codelengths (Section 3.3), and (3) the computational efficiency of local bits-back coding for use in practice. We focused on Flow++ [19], a recently proposed RealNVP-type flow with a flow-based dequantizer. We used all concepts presented in this paper: Algorithm 1 for elementwise and convolution flows [23], Algorithm 2 for coupling layers, the compositional method of Section 3.4.3, and Algorithm 3 for dequantization. We used asymmetric numeral systems (ANS) [12], following the BB-ANS [42] and Bit-Swap [26] algorithms for VAEs (though the ideas behind our algorithms do not depend on ANS). We expect our implementation to easily extend to other models, like flows for video [28] and audio [35], though we leave that for future work. We provide open-source code at `https://github.com/hojonathanho/localbitsback`.

**Codelengths** Table 1 lists the local bits-back codelengths on the test sets of CIFAR10, 32x32 ImageNet, and 64x64 ImageNet. The listed theoretical codelengths are the average negative log likelihoods of our model reimplementations, without importance sampling for the variational dequantization bound, and we find that our coding algorithm attains very similar lengths. To the best of our knowledge, these results are state-of-the-art for lossless compression with fully parallelizable compression and decompression when auxiliary bits are available for bits-back coding.

Table 1: Local bits-back codelengths (in bits per dimension)

| Compression algorithm | CIFAR10 | ImageNet 32x32 | ImageNet 64x64 |
|---|---|---|---|
| Theoretical | 3.116 | 3.871 | 3.701 |
| Local bits-back (ours) | 3.118 | 3.875 | 3.703 |

**Effects of precision and noise** Recall from Section 3.3 that the noise level $\sigma$ should be small to attain accurate codelengths. This means that the discretization volumes $\delta_x$ and $\delta_z$ should be small as well to make discretization effects negligible, at the expense of a larger requirement of auxiliary bits, which are not counted into bits-back codelengths [18]. Above, we fixed $\delta_x = \delta_z = 2^{-32}$ and $\sigma = 2^{-14}$, but here, we study the impact of varying $\delta = \delta_x = \delta_z$ and $\sigma$: on each dataset, we compressed 20 random datapoints in sequence, then calculated the local bits-back codelength and the auxiliary bits requirement; we did this for 5 random seeds and averaged the results. See Fig. 1 for CIFAR results, and see Appendix C for results on all models with standard deviation bars.

We indeed find that as $\delta$ and $\sigma$ decrease, the codelength becomes more accurate, and we find a sharp transition in performance when $\delta$ is too large relative to $\sigma$, indicating that coarse discretization destroys noise with small scale. Also, as expected, we find that the auxiliary bits requirement grows as $\delta$ shrinks. If auxiliary bits are not available, they must be counted into the codelength for the first datapoint, which can make our method impractical when coding few datapoints or when no pre-transmitted random bits are present [42, 26]. The cost can be made negligible by coding long sequences, such as entire test sets or audio or video with large numbers of frames [35, 28].

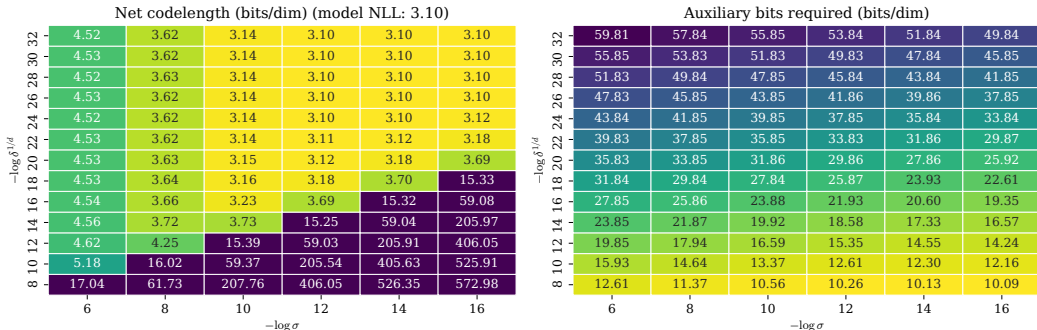

Figure 1: Effects of precision and noise parameters $\delta$ and $\sigma$ on coding a random subset of CIFAR10

**Computational efficiency** We used OpenMP-based CPU code for compression with parallel ANS streams [14], with neural net operations running on a GPU. See Table 2 for encoding timings (decoding timings in Appendix C are nearly identical), averaged over 5 runs, on 16 CPU cores and 1

Titan X GPU. We computed total CPU and GPU time for the black box algorithm (Algorithm 1) and the compositional algorithm (Section 3.4.3) on single datapoints, and we also timed the latter with batches of datapoints, made possible by its low memory requirements (this was not possible with the black box algorithm, which already needs batching to compute the Jacobian for one datapoint). We find that the total CPU and GPU time for the compositional algorithm is only slightly slower than running a pass of the flow model without coding, whereas the black box algorithm is significantly slower due to Jacobian computation. This confirms that our Jacobian-free coding techniques are crucial for practical use.

Table 2: Encoding time (in seconds per datapoint). Decoding times are nearly identical (Appendix C)

| Algorithm | Batch size | CIFAR10 | ImageNet 32x32 | ImageNet 64x64 |
|---|---|---|---|---|
| Black box (Algorithm 1) | 1 | $64.37 \pm 1.05$ | $534.74 \pm 5.91$ | $1349.65 \pm 2.30$ |
| Compositional (Section 3.4.3) | 1 | $0.77 \pm 0.01$ | $0.93 \pm 0.02$ | $0.69 \pm 0.02$ |
|  | 64 | $0.09 \pm 0.00$ | $0.17 \pm 0.00$ | $0.18 \pm 0.00$ |
| Neural net only, without coding | 1 | $0.50 \pm 0.03$ | $0.76 \pm 0.00$ | $0.44 \pm 0.00$ |
|  | 64 | $0.04 \pm 0.00$ | $0.13 \pm 0.00$ | $0.05 \pm 0.00$ |

## 5 Related work

We have built upon bits-back coding [47, 18, 38, 13, 12, 42, 26] to enable flow models to perform lossless compression, which is already possible with VAEs and autoregressive models with certain tradeoffs. VAEs and flow models (RealNVP-type models specifically) currently attain similar theoretical codelengths on image datasets [19, 29] and have similarly fast coding algorithms, but VAEs are more difficult to train due to posterior collapse [4], which implies worse net codelengths unless carefully tuned by the practitioner. Compared with the most recent instantiation of bits-back coding for hierarchical VAEs [26], our algorithm and models attain better net codelengths at the expense of a large number of auxiliary bits: on 32x32 ImageNet, we attain a net codelength of 3.88 bits/dim at the expense of approximately 40 bits/dim of auxiliary bits (depending on hyperparameter settings), but the VAEs can attain a net codelength of 4.48 bits/dim with only approximately 2.5 bits/dim of auxiliary bits [26]. As discussed in Section 4, auxiliary bits pose a problem when coding a few datapoints at a time, but not when coding long sequences like entire test sets or long videos. It would be interesting to examine the compression performance of models which are VAE-flow hybrids, of which dequantized flows are a special case (Section 3.5).

Meanwhile, autoregressive models currently attain the best codelengths (2.80 bits/dim on CIFAR10 and 3.44 bits/dim on ImageNet 64x64 [6]), but decoding, just like sampling, is extremely slow due to serial model evaluations. Both our compositional algorithm for RealNVP-type flows and algorithms for VAEs built from independent distributions are parallelizable over data dimensions and use a single model pass for both encoding and decoding.

Concurrent work [17] proposes Eq. (6) and its analysis in Appendix A to connect flows with VAEs to design new types of generative models, while by contrast, we take a pretrained, off-the-shelf flow model and employ Eq. (6) as artificial noise for compression. While the local bits-back coding concept and the black-box Algorithm 1 work for any flow, our fast linear time coding algorithms are specialized to autoregressive flows and the RealNVP family; it would be interesting to find fast coding algorithms for other types of flows [15, 3], investigate non-image modalities [28, 35], and explore connections with other literature on compression with neural networks [1, 2, 16, 40, 37].

## 6 Conclusion

We presented local bits-back coding, a technique for designing lossless compression algorithms backed by flow models. Along with a compression interpretation of dequantization, we presented concrete coding algorithms for various types of flows, culminating in an algorithm for RealNVP-type models that is fully parallelizable for encoding and decoding, runs in linear time and space, and achieves codelengths very close to theoretical predictions on high-dimensional data. As modern flow models attain excellent theoretical codelengths via straightforward, stable training, we hope that they will become viable for practical compression with the help of our algorithms, and more broadly, we hope that our work will open up new possibilities for deep generative models in compression.

**Acknowledgments**

We thank Peter Chen and Stas Tiomkin for discussions and feedback. This work was supported by a UC Berkeley EECS Departmental Fellowship.

## Footnotes

[1]We always use base 2 logarithms.

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
