[Supplementary Material]

# A Details on local bits-back coding

Here, we show that the expected codelength of local bits-back coding agrees with Eq. (5) up to first order:

$$\mathbb{E}_{\mathbf{z}} L(\mathbf{x}, \mathbf{z}) = -\log p(\mathbf{x}) \delta_x + O(\sigma^2) \tag{13}$$

Sufficient conditions for the following argument are that the prior log density and the inverse of the flow have bounded derivatives of all orders. Let $\mathbf{y} = f(\mathbf{x})$ and let $\mathbf{J}$ be the Jacobian of $f$ at $\mathbf{x}$. If we write $\mathbf{z} = \mathbf{y} + \sigma \mathbf{J} \boldsymbol{\epsilon}$ for $\boldsymbol{\epsilon} \sim \mathcal{N}(\mathbf{0}, \mathbf{I})$, the local bits-back codelength satisfies:

$$\mathbb{E}_{\mathbf{z}} L(\mathbf{x}, \mathbf{z}) + \log \delta_x = \mathbb{E}_{\boldsymbol{\epsilon}} L(\mathbf{x}, \mathbf{y} + \sigma \mathbf{J} \boldsymbol{\epsilon}) + \log \delta_x \tag{14}$$

$$= \underbrace{\mathbb{E}_{\boldsymbol{\epsilon}} \log \mathcal{N}(\mathbf{y} + \sigma \mathbf{J} \boldsymbol{\epsilon}; \mathbf{y}, \sigma^2 \mathbf{J} \mathbf{J}^\top)}_{(a)} - \underbrace{\mathbb{E}_{\boldsymbol{\epsilon}} \log \mathcal{N}(\mathbf{x}; f^{-1}(\mathbf{y} + \sigma \mathbf{J} \boldsymbol{\epsilon}), \sigma^2 \mathbf{I})}_{(b)} - \underbrace{\mathbb{E}_{\boldsymbol{\epsilon}} \log p(\mathbf{y} + \sigma \mathbf{J} \boldsymbol{\epsilon})}_{(c)}$$

We proceed by calculating each term. The first term (a) is the negative differential entropy of a Gaussian with covariance matrix $\sigma^2 \mathbf{J} \mathbf{J}^\top$:

$$\mathbb{E}_{\boldsymbol{\epsilon}} \log \mathcal{N}(\sigma \mathbf{J} \boldsymbol{\epsilon}; \mathbf{0}, \sigma^2 \mathbf{J} \mathbf{J}^\top) = -\frac{d}{2} \log(2\pi e \sigma^2) - \log |\det \mathbf{J}| \tag{15}$$

We calculate the second term (b) by taking a Taylor expansion of $f^{-1}$ around $\mathbf{y}$. Let $f_i^{-1}$ denote the $i^{\text{th}}$ coordinate of $f^{-1}$. The inverse function theorem yields

$$f_i^{-1}(\mathbf{y} + \sigma \mathbf{J} \boldsymbol{\epsilon}) = f_i^{-1}(\mathbf{y}) + \nabla f_i^{-1}(\mathbf{y})^\top (\sigma \mathbf{J} \boldsymbol{\epsilon}) + \frac{1}{2}(\sigma \mathbf{J} \boldsymbol{\epsilon})^\top \nabla^2 f_i^{-1}(\mathbf{y})(\sigma \mathbf{J} \boldsymbol{\epsilon}) + O(\sigma^3) \tag{16}$$

$$= x_i + \sigma \epsilon_i + \frac{\sigma^2}{2} \boldsymbol{\epsilon}^\top \mathbf{M}_i \boldsymbol{\epsilon} + O(\sigma^3) \tag{17}$$

where $\mathbf{M}_i := \mathbf{J}^\top \nabla^2 f_i^{-1}(\mathbf{y}) \mathbf{J}$. Write $\mathbf{v}_{\boldsymbol{\epsilon}} := [\boldsymbol{\epsilon}^\top \mathbf{M}_1 \boldsymbol{\epsilon} \quad \cdots \quad \boldsymbol{\epsilon}^\top \mathbf{M}_d \boldsymbol{\epsilon}]^\top$, so that the previous equation can be written in vector form as $f^{-1}(\mathbf{y} + \sigma \mathbf{J} \boldsymbol{\epsilon}) = \mathbf{x} + \sigma \boldsymbol{\epsilon} + \frac{\sigma^2}{2} \mathbf{v}_{\boldsymbol{\epsilon}} + O(\sigma^3)$. With this in hand, term (b) reduces to:

$$-\mathbb{E}_{\boldsymbol{\epsilon}} \log \mathcal{N}(\mathbf{x}; f^{-1}(\mathbf{y} + \sigma \mathbf{J} \boldsymbol{\epsilon}), \sigma^2 \mathbf{I}) = -\mathbb{E}_{\boldsymbol{\epsilon}} \log \mathcal{N}\left(\mathbf{x}; \mathbf{x} + \sigma \boldsymbol{\epsilon} + \frac{\sigma^2}{2} \mathbf{v}_{\boldsymbol{\epsilon}} + O(\sigma^3), \sigma^2 \mathbf{I}\right) \tag{18}$$

$$= \mathbb{E}_{\boldsymbol{\epsilon}} \left[ \frac{d}{2} \log(2\pi\sigma^2) + \frac{\log e}{2\sigma^2} \left( \|\sigma \boldsymbol{\epsilon}\|^2 + \sigma^3 \boldsymbol{\epsilon}^\top \mathbf{v}_{\boldsymbol{\epsilon}} + O(\sigma^4) \right) \right] \tag{19}$$

$$= \frac{d}{2} \log(2\pi e \sigma^2) + \frac{\sigma \log e}{2} \mathbb{E}_{\boldsymbol{\epsilon}} \left[ \boldsymbol{\epsilon}^\top \mathbf{v}_{\boldsymbol{\epsilon}} \right] + O(\sigma^2) \tag{20}$$

Because the coordinates of $\boldsymbol{\epsilon}$ are independent and have zero third moment, we have

$$\mathbb{E}_{\boldsymbol{\epsilon}} \left[ \boldsymbol{\epsilon}^\top \mathbf{v}_{\boldsymbol{\epsilon}} \right] = \mathbb{E}_{\boldsymbol{\epsilon}} \left[ \sum_i \epsilon_i \boldsymbol{\epsilon}^\top \mathbf{M}_i \boldsymbol{\epsilon} \right] = \mathbb{E}_{\boldsymbol{\epsilon}} \left[ \sum_{i,j,k} (\mathbf{M}_i)_{jk} \epsilon_i \epsilon_j \epsilon_k \right] = \sum_{i,j,k} (\mathbf{M}_i)_{jk} \mathbb{E}_{\boldsymbol{\epsilon}} \left[ \epsilon_i \epsilon_j \epsilon_k \right] = 0 \tag{21}$$

which implies that

$$-\mathbb{E}_{\boldsymbol{\epsilon}} \log \mathcal{N}(\mathbf{x}; f^{-1}(\mathbf{y} + \sigma \mathbf{J} \boldsymbol{\epsilon}), \sigma^2 \mathbf{I}) = \frac{d}{2} \log(2\pi e \sigma^2) + O(\sigma^2) \tag{22}$$

The final term (c) is given by

$$-\mathbb{E}_{\boldsymbol{\epsilon}} \log p(\mathbf{y} + \sigma \mathbf{J} \boldsymbol{\epsilon}) = -\mathbb{E}_{\boldsymbol{\epsilon}} \left[ \log p(\mathbf{y}) + \nabla \log p(\mathbf{y})^\top (\sigma \mathbf{J} \boldsymbol{\epsilon}) + O(\sigma^2) \right] \tag{23}$$

$$= -\log p(\mathbf{y}) - (\nabla \log p(\mathbf{y})^\top \sigma \mathbf{J}) \mathbb{E}_{\boldsymbol{\epsilon}} \boldsymbol{\epsilon} + O(\sigma^2) \tag{24}$$

$$= -\log p(\mathbf{y}) + O(\sigma^2) \tag{25}$$

Altogether, summing Eqs. (15), (22) and (25) yields the total codelength

$$\mathbb{E}_{\mathbf{z}} L(\mathbf{x}, \mathbf{z}) = -\log p(\mathbf{y}) - \log |\det \mathbf{J}| - \log \delta_x + O(\sigma^2) \tag{26}$$

which, to first order, does not depend on $\sigma$, and matches Eq. (5).

# B  Full algorithms

This appendix lists the full pseudocode of our coding algorithms including decoding procedures, which we omitted from the main text for brevity.

---

**Algorithm 1** Local bits-back coding: for black box flows

---

**Require:** flow $f$, discretization volumes $\delta_x$, $\delta_z$, noise level $\sigma$
1: **procedure** ENCODE($\bar{\mathbf{x}}$)
2:     $\mathbf{J} \leftarrow \mathbf{J}_f(\bar{\mathbf{x}})$                                           ▷ Compute the Jacobian of $f$ at $\bar{\mathbf{x}}$
3:     Decode $\bar{\mathbf{z}} \sim \mathcal{N}(f(\bar{\mathbf{x}}), \sigma^2 \mathbf{J}\mathbf{J}^\top)\, \delta_z$                      ▷ By converting to an AR model (Section 3.4.1)
4:     Encode $\bar{\mathbf{x}}$ using $\mathcal{N}(f^{-1}(\bar{\mathbf{z}}), \sigma^2 \mathbf{I})\, \delta_x$
5:     Encode $\bar{\mathbf{z}}$ using $p(\bar{\mathbf{z}})\, \delta_z$
6: **end procedure**

7: **procedure** DECODE( )
8:     Decode $\bar{\mathbf{z}} \sim p(\bar{\mathbf{z}})\, \delta_z$
9:     Decode $\bar{\mathbf{x}} \sim \mathcal{N}(f^{-1}(\bar{\mathbf{z}}), \sigma^2 \mathbf{I})\, \delta_x$
10:    $\mathbf{J} \leftarrow \mathbf{J}_f(\bar{\mathbf{x}})$                                          ▷ Compute the Jacobian of $f$ at $\bar{\mathbf{x}}$
11:    Encode $\bar{\mathbf{z}}$ using $\mathcal{N}(f(\bar{\mathbf{x}}), \sigma^2 \mathbf{J}\mathbf{J}^\top)\, \delta_z$               ▷ By converting to an AR model (Section 3.4.1)
12:    **return** $\bar{\mathbf{x}}$
13: **end procedure**

---

**Algorithm 2** Local bits-back coding: for autoregressive flows

---

**Require:** autoregressive flow $f$, discretization volumes $\delta_x$, $\delta_z$, noise level $\sigma$
1: **procedure** ENCODE($\bar{\mathbf{x}}$)
2:     **for** $i = d, \ldots, 1$ **do**                        ▷ Iteration ordering not mandatory, but convenient for ANS
3:         Decode $\bar{z}_i \sim \mathcal{N}(f_i(\bar{x}_i; \bar{\mathbf{x}}_{<i}), (\sigma f_i'(\bar{x}_i; \bar{\mathbf{x}}_{<i}))^2)\, \delta_z^{1/d}$   ▷ Neural net operations parallelizable over $i$
4:         Encode $\bar{x}_i$ using $\mathcal{N}(f_i^{-1}(\bar{z}_i; \bar{\mathbf{x}}_{<i}), \sigma^2)\, \delta_x^{1/d}$
5:     **end for**
6:     Encode $\bar{\mathbf{z}}$ using $p(\bar{\mathbf{z}})\, \delta_z$
7: **end procedure**

8: **procedure** DECODE( )
9:     Decode $\bar{\mathbf{z}} \sim p(\bar{\mathbf{z}})\, \delta_z$
10:    **for** $i = 1, \ldots, d$ **do**                    ▷ Order should be the opposite of encoding when using ANS
11:        Decode $\bar{x}_i \sim \mathcal{N}(f_i^{-1}(\bar{z}_i; \bar{\mathbf{x}}_{<i}), \sigma^2)\, \delta_x^{1/d}$
12:        Encode $\bar{z}_i$ using $\mathcal{N}(f_i(\bar{x}_i; \bar{\mathbf{x}}_{<i}), (\sigma f_i'(\bar{x}_i; \bar{\mathbf{x}}_{<i}))^2)\, \delta_z^{1/d}$
13:    **end for**
14:    **return** $\bar{\mathbf{x}}$
15: **end procedure**

---

**Algorithm 2** Local bits-back coding: for autoregressive flows, specialized to coupling layers

---

**Require:** coupling layer $f$, discretization volumes $\delta_x, \delta_z$, noise level $\sigma$
$\qquad$ $f$ has the form $\mathbf{z}_{\le d/2} = \mathbf{x}_{\le d/2}, \mathbf{z}_{>d/2} = f(\mathbf{x}_{>d/2}; \mathbf{x}_{\le d/2})$, where $f(\,\cdot\,; \mathbf{x}_{\le d/2})$ operates elementwise

1: **procedure** ENCODE($\bar{\mathbf{x}}$)
2: $\quad$ **for** $i = d, \ldots, d/2 + 1$ **do** $\hfill$ ▷ Neural net operations parallelizable over $i$
3: $\qquad$ Decode $\bar{z}_i \sim \mathcal{N}(f_i(\bar{x}_i; \bar{\mathbf{x}}_{\le d/2}), (\sigma f_i'(\bar{x}_i; \bar{\mathbf{x}}_{\le d/2}))^2)\,\delta_z^{1/d}$
4: $\qquad$ Encode $\bar{x}_i$ using $\mathcal{N}(f_i^{-1}(\bar{z}_i; \bar{\mathbf{x}}_{\le d/2}), \sigma^2)\,\delta_x^{1/d}$
5: $\quad$ **end for**
6: $\quad$ **for** $i = d/2, \ldots, 1$ **do**
7: $\qquad$ $\bar{z}_i \leftarrow \bar{x}_i$
8: $\quad$ **end for**
9: $\quad$ Encode $\bar{\mathbf{z}}$ using $p(\bar{\mathbf{z}})\,\delta_z$
10: **end procedure**

11: **procedure** DECODE( )
12: $\quad$ Decode $\bar{\mathbf{z}} \sim p(\bar{\mathbf{z}})\,\delta_z$
13: $\quad$ **for** $i = 1, \ldots, d/2$ **do**
14: $\qquad$ $\bar{x}_i \leftarrow \bar{z}_i$
15: $\quad$ **end for**
16: $\quad$ **for** $i = d/2 + 1, \ldots, d$ **do** $\hfill$ ▷ Neural net operations parallelizable over $i$
17: $\qquad$ Decode $\bar{x}_i \sim \mathcal{N}(f_i^{-1}(\bar{z}_i; \bar{\mathbf{x}}_{\le d/2}), \sigma^2)\,\delta_x^{1/d}$
18: $\qquad$ Encode $\bar{z}_i$ using $\mathcal{N}(f_i(\bar{x}_i; \bar{\mathbf{x}}_{\le d/2}), (\sigma f_i'(\bar{x}_i; \bar{\mathbf{x}}_{\le d/2}))^2)\,\delta_z^{1/d}$
19: $\quad$ **end for**
20: $\quad$ **return** $\bar{\mathbf{x}}$
21: **end procedure**

---

**Algorithm 3** Local bits-back coding with variational dequantization

---

**Require:** flow density $p$, dequantization flow conditional density $q$, discretization volume $\delta_x$
1: **procedure** ENCODE($\mathbf{x}^\circ$) $\hfill$ ▷ $\mathbf{x}^\circ$ is discrete data
2: $\quad$ Decode $\bar{\mathbf{u}} \sim q(\bar{\mathbf{u}}|\mathbf{x}^\circ)\,\delta_x$ via local bits-back coding
3: $\quad$ $\bar{\mathbf{x}} \leftarrow \mathbf{x}^\circ + \bar{\mathbf{u}}$ $\hfill$ ▷ Dequantize
4: $\quad$ Encode $\bar{\mathbf{x}}$ using $p(\bar{\mathbf{x}})\,\delta_x$ via local bits-back coding
5: **end procedure**

6: **procedure** DECODE( )
7: $\quad$ Decode $\bar{\mathbf{x}} \sim p(\bar{\mathbf{x}})\,\delta_x$ via local bits-back coding
8: $\quad$ $\mathbf{x}^\circ \leftarrow \lfloor \bar{\mathbf{x}} \rfloor$ $\hfill$ ▷ Quantize
9: $\quad$ $\bar{\mathbf{u}} \leftarrow \bar{\mathbf{x}} - \mathbf{x}^\circ$
10: $\quad$ Encode $\bar{\mathbf{u}}$ using $q(\bar{\mathbf{u}}|\mathbf{x}^\circ)\,\delta_x$ via local bits-back coding
11: $\quad$ **return** $\mathbf{x}^\circ$
12: **end procedure**

---

# C Experiment details

Figure 2 and Tables 3 to 5 show complete results for the experiments in Section 4, which examine how compression performance is affected by the precision and noise level parameters $\delta$ and $\sigma$. Table 6 contains timing results for decoding.

Figure 2: Codelengths on subsets of CIFAR10 (top), ImageNet 32x32 (middle), and ImageNet 64x64 (bottom)

## Table 3: Codelengths on subset of CIFAR10 (bits/dim)

| | $\sigma = 2^{-6}$ | $\sigma = 2^{-8}$ | $\sigma = 2^{-10}$ | $\sigma = 2^{-12}$ | $\sigma = 2^{-14}$ | $\sigma = 2^{-16}$ |
|---|---|---|---|---|---|---|
| **Net codelength** | | | | | | |
| $\delta^{1/d} = 2^{-32}$ | $4.520 \pm 0.082$ | $3.623 \pm 0.109$ | $3.141 \pm 0.138$ | $3.102 \pm 0.141$ | $3.099 \pm 0.140$ | $3.099 \pm 0.140$ |
| $\delta^{1/d} = 2^{-30}$ | $4.526 \pm 0.082$ | $3.624 \pm 0.108$ | $3.141 \pm 0.137$ | $3.103 \pm 0.141$ | $3.099 \pm 0.141$ | $3.099 \pm 0.142$ |
| $\delta^{1/d} = 2^{-28}$ | $4.519 \pm 0.081$ | $3.628 \pm 0.110$ | $3.142 \pm 0.138$ | $3.103 \pm 0.141$ | $3.099 \pm 0.144$ | $3.099 \pm 0.142$ |
| $\delta^{1/d} = 2^{-26}$ | $4.528 \pm 0.083$ | $3.624 \pm 0.107$ | $3.141 \pm 0.138$ | $3.101 \pm 0.140$ | $3.098 \pm 0.141$ | $3.104 \pm 0.143$ |
| $\delta^{1/d} = 2^{-24}$ | $4.525 \pm 0.075$ | $3.625 \pm 0.111$ | $3.139 \pm 0.134$ | $3.102 \pm 0.143$ | $3.103 \pm 0.144$ | $3.119 \pm 0.144$ |
| $\delta^{1/d} = 2^{-22}$ | $4.530 \pm 0.085$ | $3.624 \pm 0.112$ | $3.142 \pm 0.134$ | $3.107 \pm 0.140$ | $3.119 \pm 0.142$ | $3.181 \pm 0.146$ |
| $\delta^{1/d} = 2^{-20}$ | $4.528 \pm 0.081$ | $3.634 \pm 0.103$ | $3.147 \pm 0.135$ | $3.122 \pm 0.141$ | $3.178 \pm 0.137$ | $3.691 \pm 0.160$ |
| $\delta^{1/d} = 2^{-18}$ | $4.529 \pm 0.077$ | $3.639 \pm 0.103$ | $3.163 \pm 0.138$ | $3.181 \pm 0.141$ | $3.698 \pm 0.149$ | $15.333 \pm 0.387$ |
| $\delta^{1/d} = 2^{-16}$ | $4.536 \pm 0.081$ | $3.655 \pm 0.102$ | $3.228 \pm 0.143$ | $3.692 \pm 0.140$ | $15.323 \pm 0.433$ | $59.078 \pm 0.897$ |
| $\delta^{1/d} = 2^{-14}$ | $4.558 \pm 0.081$ | $3.716 \pm 0.104$ | $3.732 \pm 0.148$ | $15.252 \pm 0.448$ | $59.042 \pm 0.926$ | $205.973 \pm 2.394$ |
| $\delta^{1/d} = 2^{-12}$ | $4.622 \pm 0.078$ | $4.252 \pm 0.108$ | $15.389 \pm 0.361$ | $59.031 \pm 0.979$ | $205.908 \pm 2.238$ | $406.046 \pm 1.863$ |
| $\delta^{1/d} = 2^{-10}$ | $5.179 \pm 0.080$ | $16.015 \pm 0.347$ | $59.370 \pm 0.988$ | $205.539 \pm 2.159$ | $405.630 \pm 1.920$ | $525.914 \pm 1.951$ |
| $\delta^{1/d} = 2^{-8}$ | $17.040 \pm 0.332$ | $61.730 \pm 0.892$ | $207.756 \pm 2.065$ | $406.051 \pm 1.772$ | $526.353 \pm 1.720$ | $572.980 \pm 1.416$ |
| **Auxiliary bits required** | | | | | | |
| $\delta^{1/d} = 2^{-32}$ | $59.813 \pm 0.078$ | $57.836 \pm 0.063$ | $55.847 \pm 0.072$ | $53.844 \pm 0.078$ | $51.840 \pm 0.088$ | $49.844 \pm 0.070$ |
| $\delta^{1/d} = 2^{-30}$ | $55.846 \pm 0.076$ | $53.833 \pm 0.081$ | $51.830 \pm 0.086$ | $49.829 \pm 0.094$ | $47.843 \pm 0.085$ | $45.854 \pm 0.079$ |
| $\delta^{1/d} = 2^{-28}$ | $51.833 \pm 0.079$ | $49.841 \pm 0.072$ | $47.846 \pm 0.073$ | $45.844 \pm 0.074$ | $43.844 \pm 0.082$ | $41.848 \pm 0.076$ |
| $\delta^{1/d} = 2^{-26}$ | $47.831 \pm 0.087$ | $45.847 \pm 0.082$ | $43.855 \pm 0.080$ | $41.861 \pm 0.084$ | $39.858 \pm 0.076$ | $37.846 \pm 0.078$ |
| $\delta^{1/d} = 2^{-24}$ | $43.841 \pm 0.060$ | $41.849 \pm 0.068$ | $39.853 \pm 0.080$ | $37.855 \pm 0.083$ | $35.838 \pm 0.065$ | $33.844 \pm 0.067$ |
| $\delta^{1/d} = 2^{-22}$ | $39.832 \pm 0.101$ | $37.848 \pm 0.072$ | $35.848 \pm 0.069$ | $33.834 \pm 0.068$ | $31.858 \pm 0.060$ | $29.874 \pm 0.087$ |
| $\delta^{1/d} = 2^{-20}$ | $35.834 \pm 0.064$ | $33.850 \pm 0.082$ | $31.857 \pm 0.086$ | $29.859 \pm 0.082$ | $27.861 \pm 0.093$ | $25.923 \pm 0.060$ |
| $\delta^{1/d} = 2^{-18}$ | $31.840 \pm 0.075$ | $29.845 \pm 0.081$ | $27.845 \pm 0.072$ | $25.874 \pm 0.069$ | $23.932 \pm 0.086$ | $22.608 \pm 0.108$ |
| $\delta^{1/d} = 2^{-16}$ | $27.852 \pm 0.090$ | $25.856 \pm 0.074$ | $23.875 \pm 0.084$ | $21.931 \pm 0.072$ | $20.595 \pm 0.107$ | $19.350 \pm 0.000$ |
| $\delta^{1/d} = 2^{-14}$ | $23.852 \pm 0.070$ | $21.867 \pm 0.073$ | $19.918 \pm 0.075$ | $18.578 \pm 0.108$ | $17.331 \pm 0.000$ | $16.573 \pm 0.000$ |
| $\delta^{1/d} = 2^{-12}$ | $19.853 \pm 0.073$ | $17.940 \pm 0.077$ | $16.590 \pm 0.114$ | $15.350 \pm 0.000$ | $14.549 \pm 0.000$ | $14.236 \pm 0.000$ |
| $\delta^{1/d} = 2^{-10}$ | $15.933 \pm 0.070$ | $14.638 \pm 0.102$ | $13.368 \pm 0.000$ | $12.610 \pm 0.000$ | $12.297 \pm 0.000$ | $12.156 \pm 0.000$ |
| $\delta^{1/d} = 2^{-8}$ | $12.607 \pm 0.108$ | $11.369 \pm 0.000$ | $10.562 \pm 0.000$ | $10.264 \pm 0.000$ | $10.134 \pm 0.000$ | $10.087 \pm 0.000$ |

## Table 4: Codelengths on subset of ImageNet 32x32 (bits/dim)

| | $\sigma = 2^{-6}$ | $\sigma = 2^{-8}$ | $\sigma = 2^{-10}$ | $\sigma = 2^{-12}$ | $\sigma = 2^{-14}$ | $\sigma = 2^{-16}$ |
|---|---|---|---|---|---|---|
| **Net codelength** | | | | | | |
| $\delta^{1/d} = 2^{-32}$ | $4.513 \pm 0.050$ | $3.961 \pm 0.070$ | $3.839 \pm 0.086$ | $3.825 \pm 0.091$ | $3.825 \pm 0.091$ | $3.831 \pm 0.091$ |
| $\delta^{1/d} = 2^{-30}$ | $4.513 \pm 0.047$ | $3.962 \pm 0.069$ | $3.839 \pm 0.086$ | $3.826 \pm 0.091$ | $3.833 \pm 0.092$ | $3.854 \pm 0.089$ |
| $\delta^{1/d} = 2^{-28}$ | $4.517 \pm 0.052$ | $3.963 \pm 0.071$ | $3.839 \pm 0.088$ | $3.833 \pm 0.092$ | $3.850 \pm 0.090$ | $3.917 \pm 0.090$ |
| $\delta^{1/d} = 2^{-26}$ | $4.522 \pm 0.049$ | $3.965 \pm 0.069$ | $3.849 \pm 0.087$ | $3.852 \pm 0.090$ | $3.925 \pm 0.090$ | $4.166 \pm 0.089$ |
| $\delta^{1/d} = 2^{-24}$ | $4.528 \pm 0.050$ | $3.973 \pm 0.071$ | $3.871 \pm 0.087$ | $3.933 \pm 0.095$ | $4.148 \pm 0.091$ | $4.763 \pm 0.101$ |
| $\delta^{1/d} = 2^{-22}$ | $4.538 \pm 0.048$ | $3.995 \pm 0.071$ | $3.947 \pm 0.089$ | $4.151 \pm 0.095$ | $4.769 \pm 0.097$ | $6.437 \pm 0.110$ |
| $\delta^{1/d} = 2^{-20}$ | $4.569 \pm 0.048$ | $4.070 \pm 0.074$ | $4.173 \pm 0.076$ | $4.752 \pm 0.087$ | $6.434 \pm 0.143$ | $11.231 \pm 0.303$ |
| $\delta^{1/d} = 2^{-18}$ | $4.653 \pm 0.044$ | $4.292 \pm 0.072$ | $4.781 \pm 0.086$ | $6.452 \pm 0.136$ | $11.194 \pm 0.305$ | $33.878 \pm 0.704$ |
| $\delta^{1/d} = 2^{-16}$ | $4.895 \pm 0.041$ | $4.889 \pm 0.054$ | $6.427 \pm 0.082$ | $11.148 \pm 0.314$ | $33.878 \pm 0.858$ | $117.029 \pm 1.249$ |
| $\delta^{1/d} = 2^{-14}$ | $5.524 \pm 0.044$ | $6.524 \pm 0.106$ | $11.119 \pm 0.359$ | $33.883 \pm 0.855$ | $117.121 \pm 1.431$ | $332.916 \pm 2.599$ |
| $\delta^{1/d} = 2^{-12}$ | $7.230 \pm 0.094$ | $11.098 \pm 0.248$ | $33.831 \pm 0.755$ | $116.947 \pm 1.200$ | $332.488 \pm 2.464$ | $540.396 \pm 2.686$ |
| $\delta^{1/d} = 2^{-10}$ | $11.700 \pm 0.317$ | $33.523 \pm 0.891$ | $116.809 \pm 1.177$ | $332.633 \pm 2.736$ | $540.065 \pm 2.580$ | $609.042 \pm 2.046$ |
| $\delta^{1/d} = 2^{-8}$ | $33.709 \pm 0.746$ | $116.615 \pm 1.286$ | $333.066 \pm 2.688$ | $540.738 \pm 2.327$ | $609.349 \pm 1.831$ | $633.963 \pm 1.950$ |
| **Auxiliary bits required** | | | | | | |
| $\delta^{1/d} = 2^{-32}$ | $59.996 \pm 0.083$ | $57.996 \pm 0.066$ | $55.977 \pm 0.065$ | $53.986 \pm 0.067$ | $51.981 \pm 0.061$ | $49.975 \pm 0.057$ |
| $\delta^{1/d} = 2^{-30}$ | $55.988 \pm 0.074$ | $53.984 \pm 0.066$ | $52.000 \pm 0.062$ | $49.973 \pm 0.071$ | $47.982 \pm 0.064$ | $45.947 \pm 0.064$ |
| $\delta^{1/d} = 2^{-28}$ | $51.984 \pm 0.084$ | $49.984 \pm 0.071$ | $47.985 \pm 0.072$ | $45.963 \pm 0.066$ | $43.950 \pm 0.069$ | $41.911 \pm 0.080$ |
| $\delta^{1/d} = 2^{-26}$ | $47.971 \pm 0.079$ | $45.976 \pm 0.075$ | $43.974 \pm 0.077$ | $41.947 \pm 0.071$ | $39.827 \pm 0.033$ | $37.537 \pm 0.039$ |
| $\delta^{1/d} = 2^{-24}$ | $43.991 \pm 0.047$ | $41.969 \pm 0.072$ | $39.934 \pm 0.076$ | $37.841 \pm 0.063$ | $35.627 \pm 0.051$ | $33.068 \pm 0.071$ |
| $\delta^{1/d} = 2^{-22}$ | $39.986 \pm 0.063$ | $37.962 \pm 0.067$ | $35.850 \pm 0.052$ | $33.582 \pm 0.059$ | $30.980 \pm 0.050$ | $29.777 \pm 0.025$ |
| $\delta^{1/d} = 2^{-20}$ | $35.938 \pm 0.072$ | $33.872 \pm 0.071$ | $31.629 \pm 0.066$ | $29.028 \pm 0.060$ | $27.787 \pm 0.021$ | $26.765 \pm 0.019$ |
| $\delta^{1/d} = 2^{-18}$ | $31.816 \pm 0.045$ | $29.633 \pm 0.045$ | $26.965 \pm 0.019$ | $25.800 \pm 0.031$ | $24.736 \pm 0.024$ | $23.446 \pm 0.044$ |
| $\delta^{1/d} = 2^{-16}$ | $27.664 \pm 0.048$ | $25.111 \pm 0.050$ | $23.781 \pm 0.010$ | $22.762 \pm 0.024$ | $21.425 \pm 0.048$ | $19.386 \pm 0.000$ |
| $\delta^{1/d} = 2^{-14}$ | $23.175 \pm 0.045$ | $21.773 \pm 0.013$ | $20.742 \pm 0.012$ | $19.462 \pm 0.015$ | $17.365 \pm 0.000$ | $16.589 \pm 0.000$ |
| $\delta^{1/d} = 2^{-12}$ | $19.775 \pm 0.029$ | $18.735 \pm 0.029$ | $17.435 \pm 0.026$ | $15.366 \pm 0.000$ | $14.582 \pm 0.000$ | $14.236 \pm 0.002$ |
| $\delta^{1/d} = 2^{-10}$ | $16.788 \pm 0.023$ | $15.435 \pm 0.023$ | $13.389 \pm 0.000$ | $12.598 \pm 0.000$ | $12.271 \pm 0.003$ | $12.152 \pm 0.002$ |
| $\delta^{1/d} = 2^{-8}$ | $13.451 \pm 0.013$ | $11.362 \pm 0.000$ | $10.586 \pm 0.000$ | $10.256 \pm 0.001$ | $10.133 \pm 0.001$ | $10.087 \pm 0.001$ |

Table 5: Codelengths on subset of ImageNet 64x64 (bits/dim)

| | $\sigma = 2^{-8}$ | $\sigma = 2^{-10}$ | $\sigma = 2^{-12}$ | $\sigma = 2^{-14}$ | $\sigma = 2^{-16}$ |
|---|---|---|---|---|---|
| **Net codelength** | | | | | |
| $\delta^{1/d} = 2^{-32}$ | $3.771 \pm 0.062$ | $3.642 \pm 0.074$ | $3.627 \pm 0.078$ | $3.626 \pm 0.078$ | $3.626 \pm 0.078$ |
| $\delta^{1/d} = 2^{-30}$ | $3.770 \pm 0.062$ | $3.642 \pm 0.073$ | $3.627 \pm 0.078$ | $3.626 \pm 0.078$ | $3.627 \pm 0.078$ |
| $\delta^{1/d} = 2^{-28}$ | $3.772 \pm 0.062$ | $3.642 \pm 0.074$ | $3.627 \pm 0.078$ | $3.627 \pm 0.078$ | $3.628 \pm 0.078$ |
| $\delta^{1/d} = 2^{-26}$ | $3.772 \pm 0.062$ | $3.642 \pm 0.074$ | $3.628 \pm 0.078$ | $3.629 \pm 0.078$ | $3.640 \pm 0.078$ |
| $\delta^{1/d} = 2^{-24}$ | $3.772 \pm 0.061$ | $3.643 \pm 0.074$ | $3.630 \pm 0.078$ | $3.641 \pm 0.079$ | $3.705 \pm 0.079$ |
| $\delta^{1/d} = 2^{-22}$ | $3.774 \pm 0.062$ | $3.645 \pm 0.074$ | $3.641 \pm 0.077$ | $3.702 \pm 0.079$ | $4.014 \pm 0.081$ |
| $\delta^{1/d} = 2^{-20}$ | $3.776 \pm 0.063$ | $3.659 \pm 0.074$ | $3.705 \pm 0.078$ | $4.017 \pm 0.080$ | $5.563 \pm 0.096$ |
| $\delta^{1/d} = 2^{-18}$ | $3.788 \pm 0.061$ | $3.721 \pm 0.077$ | $4.017 \pm 0.082$ | $5.559 \pm 0.081$ | $20.108 \pm 0.201$ |
| $\delta^{1/d} = 2^{-16}$ | $3.852 \pm 0.063$ | $4.032 \pm 0.075$ | $5.557 \pm 0.076$ | $20.128 \pm 0.183$ | $73.619 \pm 0.724$ |
| $\delta^{1/d} = 2^{-14}$ | $4.158 \pm 0.066$ | $5.571 \pm 0.079$ | $20.100 \pm 0.187$ | $73.637 \pm 0.759$ | $228.844 \pm 0.629$ |
| $\delta^{1/d} = 2^{-12}$ | $5.721 \pm 0.067$ | $20.142 \pm 0.243$ | $73.596 \pm 0.717$ | $228.707 \pm 0.651$ | $409.066 \pm 1.126$ |
| $\delta^{1/d} = 2^{-10}$ | $20.321 \pm 0.222$ | $73.654 \pm 0.676$ | $228.471 \pm 0.703$ | $408.415 \pm 0.903$ | $485.477 \pm 1.315$ |
| $\delta^{1/d} = 2^{-8}$ | $74.060 \pm 0.837$ | $228.752 \pm 0.565$ | $408.316 \pm 0.980$ | $485.631 \pm 1.070$ | $517.896 \pm 1.127$ |
| **Auxiliary bits required** | | | | | |
| $\delta^{1/d} = 2^{-32}$ | $58.001 \pm 0.046$ | $55.998 \pm 0.049$ | $53.999 \pm 0.042$ | $52.003 \pm 0.049$ | $50.001 \pm 0.042$ |
| $\delta^{1/d} = 2^{-30}$ | $53.997 \pm 0.046$ | $51.999 \pm 0.051$ | $50.001 \pm 0.048$ | $48.012 \pm 0.040$ | $46.012 \pm 0.045$ |
| $\delta^{1/d} = 2^{-28}$ | $50.009 \pm 0.049$ | $48.013 \pm 0.044$ | $46.003 \pm 0.049$ | $44.003 \pm 0.044$ | $42.002 \pm 0.048$ |
| $\delta^{1/d} = 2^{-26}$ | $46.005 \pm 0.046$ | $44.001 \pm 0.045$ | $42.006 \pm 0.048$ | $40.001 \pm 0.045$ | $38.002 \pm 0.040$ |
| $\delta^{1/d} = 2^{-24}$ | $42.007 \pm 0.044$ | $39.998 \pm 0.040$ | $38.003 \pm 0.045$ | $36.003 \pm 0.046$ | $34.006 \pm 0.042$ |
| $\delta^{1/d} = 2^{-22}$ | $38.004 \pm 0.040$ | $36.004 \pm 0.045$ | $34.004 \pm 0.049$ | $32.008 \pm 0.047$ | $30.019 \pm 0.039$ |
| $\delta^{1/d} = 2^{-20}$ | $33.999 \pm 0.044$ | $32.004 \pm 0.041$ | $30.008 \pm 0.046$ | $28.007 \pm 0.044$ | $26.020 \pm 0.041$ |
| $\delta^{1/d} = 2^{-18}$ | $30.010 \pm 0.037$ | $28.007 \pm 0.044$ | $26.013 \pm 0.035$ | $24.017 \pm 0.032$ | $22.715 \pm 0.041$ |
| $\delta^{1/d} = 2^{-16}$ | $26.020 \pm 0.048$ | $24.013 \pm 0.039$ | $22.010 \pm 0.031$ | $20.739 \pm 0.043$ | $19.158 \pm 0.000$ |
| $\delta^{1/d} = 2^{-14}$ | $22.013 \pm 0.038$ | $20.028 \pm 0.033$ | $18.714 \pm 0.047$ | $17.158 \pm 0.000$ | $16.419 \pm 0.000$ |
| $\delta^{1/d} = 2^{-12}$ | $18.012 \pm 0.033$ | $16.719 \pm 0.044$ | $15.165 \pm 0.000$ | $14.429 \pm 0.000$ | $14.156 \pm 0.000$ |
| $\delta^{1/d} = 2^{-10}$ | $14.722 \pm 0.046$ | $13.172 \pm 0.000$ | $12.428 \pm 0.000$ | $12.160 \pm 0.000$ | $12.084 \pm 0.000$ |
| $\delta^{1/d} = 2^{-8}$ | $11.157 \pm 0.000$ | $10.415 \pm 0.000$ | $10.153 \pm 0.000$ | $10.080 \pm 0.000$ | $10.056 \pm 0.000$ |

Table 6: Decoding time (in seconds per datapoint)

| Compression algorithm | Batch size | CIFAR10 | ImageNet 32x32 | ImageNet 64x64 |
|---|---|---|---|---|
| Black box (Algorithm 1) | 1 | $65.90 \pm 0.10$ | $564.42 \pm 15.26$ | $1351.04 \pm 3.31$ |
| Compositional (Section 3.4.3) | 1 | $0.78 \pm 0.02$ | $0.92 \pm 0.00$ | $0.71 \pm 0.03$ |
| | 64 | $0.09 \pm 0.00$ | $0.17 \pm 0.00$ | $0.18 \pm 0.00$ |
| Neural net only, without coding | 1 | $0.50 \pm 0.03$ | $0.76 \pm 0.00$ | $0.44 \pm 0.00$ |
| | 64 | $0.04 \pm 0.00$ | $0.13 \pm 0.00$ | $0.05 \pm 0.00$ |