[Reviews · NeurIPS 2019]

Reviewer 1



This is a nice idea, a well-written paper, and should definitely be published. It is interesting for the NeurIPS community, because it presents a way to convert essentially arbitrary flow models into lossless compression methods. It is conceptually interesting, and there are plenty of potential applications. My only criticisms are with the discussion and evaluation. First, BB-ANS suffers from significant overhead due to the requirement to send auxiliary bits. The authors don't really discuss this sufficiently, and claim that it is negligible when encoding long sequences of i.i.d. data such as video. The reported overhead is between 1000% and 1200% of the code length, which is a lot even for video (one reason is that the first “intra” frame typically contains many more bits than the subsequent “inter” frames. Hence, to amortize 1000% of the size of the first frame could take hundreds or even thousands of subsequent frames). Overall, it would be honest to concede in the discussion that this is a shortcoming (but can of course potentially be worked around). Right now, it's up to the reader to work out that the auxiliary bits could be a problem in practice. Second, regarding runtime, the authors state that “the compositional algorithm is only slightly slower than running the neural net on its own”. This may be true in terms of “wall clock” time, but what really matters is how many CPU/GPU cycles were actually utilized. This is what is of practical concern. It appears the authors here engage in a comparison of “apples to oranges”, since it’s quite plausible that the CPU has substantially more idle time for running just the flow model compared to running bits-back coding, since BB-ANS runs on the CPU. The authors need to report cumulative CPU/GPU times rather than wall clock time to be able to make a claim about computational complexity. For instance, if the flow model doesn’t utilize the CPU at all, the authors are effectively giving their method a significant amount of additional resources over the baseline. EDIT: I thank the authors for their clarifications. Based on this, I’ll maintain my initial score.

Reviewer 2



Summary: The paper looks at the problem of constructing a bits-back compression algorithm using a generative model. The introduced algorithm, local bits-back coding, takes advantage of the structure of flow-based generative models. The paper starts out by extensively introducing flow models, quantization and bits-back coding. It then introduces local bits-back coding and a concrete algorithm. Naively applying the algorithm is relatively slow, but by exploiting the structure of the Jacobian in affine coupling layers, a faster (parallelised) version of the algorithm becomes possible. The authors further show that they can recursively apply local bits-back on different components (layers) of the flow model. The paper ends with results that are indeed close to the theoretical maximum and incur minimum overhead to code (on top of the forward pass of the Flow model). Further the model is reasonably robust to hyperparameters, but requires a significant amount of auxiliary bits. Discussion: The paper is very well written, with a thorough theoretical introduction and a precise explanation of the method. The introduced algorithm, local bits-back, is novel and produces excellent results. Especially the combination with "RealNVP style" models is elegant, since so much effort went into creating components with easy to compute jacobians and local bits-back introduces a new way to capitalise on this effort. The paper contains several colloquialisms that are worth editing out, listed below (not comprehensive). The main concern with the paper is the lack of comparison with other methods. The authors argue that VAEs suffer from posterior collapse and do not provide an exact likelihood (only a lower bound), however that is not an argument to disregard comparing to coding algorithms for them (e.g. Bit-swap, BB-ANS). Especially in terms of computational complexity, ease of use, number of auxiliary bits needed, space complexity. Notes: - line 34: "con" - too informal - line 34: "in odd contrast with" -> "in contrast to", or "at odds with" - line 65: missing "to" before "train" - line 168: ', say,' - too informal -- Rebuttal -- The rebuttal did not provide reason to increase the score.

Reviewer 3



(1) Quality: Is the submission technically sound? Are claims well supported by theoretical analysis or experimental results? Is this a complete piece of work or work in progress? Are the authors careful and honest about evaluating both the strengths and weaknesses of their work? The quality of the work is good. It is interesting, it makes clear claims and it supports them with both theoretical and empirical arguments. One aspects that this paper is lacking in is motivation/context. I understand that the issue this paper addresses is coding using flow models, however, since flow models are not commonly used for compression, it should provide motivation why one might want to use them. The paper should argue why the problem that it is solving is important for either theoreticians or practitioners. If flow models are indeed competitive with VAEs, that should be demonstrated empirically. (2) Clarity: Is the submission clearly written? Is it well organized? (If not, please make constructive suggestions for improving its clarity.) Does it adequately inform the reader? (Note: a superbly written paper provides enough information for an expert reader to reproduce its results.) The paper is well written and easy to understand. I really like that each algorithm is shown in pseudo code. One part that was unclear to me is in section 3.5, why is it a problem is log delta > 1? Why is that a waste of bits? (3) Originality: Are the tasks or methods new? Is the work a novel combination of well-known techniques? Is it clear how this work differs from previous contributions? Is related work adequately cited? All of the ideas presented are novel and original to my knowledge. Compression with flows is a new research area and I am unaware of any prior works. (4) Significance: Are the results important? Are others (researchers or practitioners) likely to use the ideas or build on them? Does the submission address a difficult task in a better way than previous work? Does it advance the state of the art in a demonstrable way? Does it provide unique data, unique conclusions about existing data, or a unique theoretical or experimental approach? For the reasons stated above, it is difficult to judge the significance for practitioners. To be able to conclusively state its importance, we would have to see comparisons to existing methods. As for researchers, I do find the idea interesting and I think it could have future applications. __________________________________________________________________________________________________________________________________________________ After reading the author's reply, I decided to keep my original score.

[Author Response · NeurIPS 2019]

We thank all reviewers for their constructive and helpful comments.

Reviewer 1: We will be sure to provide a more accurate and nuanced discussion of the downsides of our auxiliary bits requirements in a revision. The particular example we had in mind for the sentence on video was the case of compressing an hour-long video: at 30 frames per second, this is 100 thousand frames, after which we expect the auxiliary bits to be negligible. The auxiliary bits are of course not negligible for shorter videos, and we will change the text to plainly describe the downside of large auxiliary bits requirements.

Reviewer 1: Regarding runtime evaluation, what we called the "wall clock time" is the sum of the GPU time and the CPU time, and the reported time to "run the neural net on its own" is the GPU time. We will correct and clarify our terminology regarding timing here.

Reviewer 1: Regarding hardware and software differences for encoder and decoder: indeed it is crucial for all computations, especially those in the flow model, to be exactly reproducible on both sides of the communication channel, otherwise we indeed do run into catastrophic error propagation. In our implementation, we set flags to force usage of deterministic CUDA kernels, and we use the same hardware for both encoding and decoding. As with other methods that rely on exactly reproducible probability models, these issues can be addressed with careful engineering.

Reviewers 2 and 3: Comparing the most modern instantiation of bits-back coding with hierarchical VAEs (Bit-Swap), our algorithm and models have better net bitrates, at the expense of a large number of auxiliary bits. Specifically, on 32x32 Imagenet, we attain a net codelength of 3.88 bits/dim at the expense of approximately 40 bits/dim of auxiliary bits (depending on hyperparmeter settings). In contrast, the models in the Bit-Swap paper attain a net codelength of 4.48 bits/dim, with only approximately 2.5 bits/dim of auxiliary bits. Indeed, as Reviewer 1 mentioned, our auxiliary bits requirement is a downside of our method for short sequences, which do not have enough timesteps to amortize out this requirement. We will revise our paper to include this discussion.

Reviewer 3: One of the motivations of our work is theoretical interest: likelihood-based models are known to optimize lossless compression rate, but it is not always clear for any one given likelihood-based model how to achieve this rate in practice. We have filled in this gap in the literature for flow models. Another motivation is practical: when running our algorithm on RealNVP-type models, encoding/decoding passes are fast and parallelizable, unlike arithmetic coding or ANS for autoregressive models, which are slow for decoding. We also attain good net codelengths, which are currently better than those in the VAE compression literature (though at the expense of a worse auxiliary bits requirement, as just discussed).

Reviewer 3: Regarding clarification of Section 3.5: coding integer-valued data at high precision is a waste of bits, because it is not necessary to specify the data at a resolution smaller than that of a unit hypercube. In the one-dimensional case, doing so is akin to storing integers with a nonzero amount of precision after the decimal point – this is a waste of bits, because those digits after the decimal point will always be zero. The image datasets we use consist of integer data, so we wish to avoid coding bins of volume less than 1.

[Meta-Review · NeurIPS 2019]

This paper received very good scores, with strong consensus in the reviews. It presents interesting ideas for doing lossless compression with flow-based models. The paper could be strengthened with more experiments/comparisons, but it still a good contribution to Neurips.